# Effect of Protoneutron Star Magnetized Envelops in Neutrino Energy Spectra

**Vladimir N. Kondratyev** [1,2,*], **Tamara D. Lobanovskaya** [2] **and Dimash B. Torekhan** [2]

1   Bogoliubov Laboratory of Theoretical Physics, Joint Institute for Nuclear Research (JINR),
    141980 Dubna, Russia
2   Nuclear Physics Department, Dubna State University, University Str., 19, 141982 Dubna, Russia;
    tamaralobanovskaya@yandex.ru (T.D.L.); dimash.1997@mail.ru (D.B.T.)
*   Correspondence: vkondrat@theor.jinr.ru

**Abstract:** The neutrino dynamics in hot and dense magnetized matter, which corresponds with protoneutron star envelopes in the core collapse supernova explosions, is considered. The kinetic equation for a neutrino phase space distribution function is obtained, taking into account inelastic scattering by nuclear particles. The transfer component in a momentum space using transport properties is studied. The energy transfer coefficient is shown to change from positive to negative values when the neutrino energy exceeds four times the matter temperature. In the vicinity of a neutrino sphere, such effects are illustrated to lead to the energy strengthening in the neutrino spectra. As this paper demonstrates, such a property is favorable for the possibility of observing supernova neutrino fluxes using Large Volume Neutrino Telescopes.

**Keywords:** supernovae; core collapse; dynamo process; magnetic field; neutrino energy spectra; magnetars

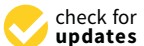



## 1. Introduction

Type II supernovae (SNe) represent one of the most energetic events, as they can, plausibly, give high energy cosmic rays, sites for synthesis of heavy nuclides (e.g., e-, s-, p-, r-process atomic nuclei), renew other nuclear components, and so on. Moreover, the SN explosion mechanism constitutes an important issue that is still under debate. In particular, the mechanism of energy transfer to all-star matter (initially bound) represents the main SN problem. Neutrino flux and/or magnetic pressures plausibly make a key contribution to explosive shock wave formation, in addition to core bounce and thermal pressure, respectively.

Being in the vicinity of neutrino sphere strong convection [1], and/or magnetorotational instability [2], can trigger a turbulent dynamo leading to a magnetic field amplification of up to tens of teratesla (TT). Prompt SN explosions can be associated with magnetic pressure contributing significant energy into the stellar matter, and bringing the predominant mechanism of shock wave formation, respectively. Another possible explosion mechanism is related to neutrino heating and reviving the stopped shock wave using the neutrinos and antineutrinos that are emitted by a cooling protoneutron star [3–5]. Neutrino absorption of nucleons gives rise to the increasing temperature and pressure of the matter behind the shock that starts to expand, pushing the shock forward. The efficiency of such a delayed CCSN mechanism depends on neutrino luminosity and the hardness of their spectra [4,5].

Since neutrinos or magnetic pressure are capable of making a significant contribution to the mechanism of supernova explosions, an analysis of neutrino transport in supernova matter, taking into account magnetic effects, represents an important issue. In addition, the possible magnetic influence on neutrino spectra are crucial for the interpretation of the r- and neutrino processes that can be also affected by magnetic fields [6–9]. As has

been recently shown [10,11], the energy exchange in neutrino–nuclear scattering can be noticeably enhanced due to the magnetic field. In the next section we consider neutrino kinetics, paying particular attention to the transfer in momentum space, on the basis of an energy transfer cross section [11]. As is illustrated in Section 3, such an approach gives a clear picture of the influence of inelastic scattering on neutrino spectra. Possibilities to observe the possible effects of this using Large Volume Neutrino Telescopes are analyzed in Section 4. Conclusions are given in Section 5.

## 2. Neutrino Kinetics

To describe neutrino kinetics, we use quite a general kinetic equation for the phase–space distribution function $f(\mathbf{r}, \mathbf{p}, l)$

$$\frac{df}{dl} = \frac{\partial f}{\partial l} + z\frac{\partial f}{\partial \mathbf{r}} + \frac{\partial \mathbf{p}}{\partial l}\frac{\partial f}{\partial \mathbf{p}} = \Lambda + \text{St}[f], \qquad (1)$$

where a distance passed by neutrino $l = c\,t$ with speed of light $c$ and time $t$, $\partial \mathbf{r}$ and $\partial \mathbf{p}$ representing the partial derivatives with respect to the spatial, $\mathbf{r}$, and momentum coordinates, $\mathbf{p} = \mathbf{z}\,E/c$, with the unit vector $\mathbf{z}$ defining the direction of neutrino momentum at energy $E$. Here, we take into account that since supernova neutrinos possess typical energies in the MeV range, much larger than the experimental rest-mass limit for active flavors <1 eV, they essentially propagate with the speed of light $c$. The momentum derivative in Equation (1), $\partial \mathbf{p}/\partial l$ accounts for an energy and momentum exchange during the neutrino collisions with environment particles. On the right hand side of Equation (1), $\Lambda$ stands for all the rates of neutrino production, absorption, and annihilation, whereas the term St[f], accounts for fluctuations in scattering processes.

## 3. Neutrino Dynamics in Magnetized Neutrino Sphere

In this work, we concentrate on neutrino kinetics in a region of neutrino decoupling from matter for dynamo active SNe. In such neutrino sphere regions, the neutrino dynamics change from diffusive to free streaming. Spectra of neutrinos emerging from a protoneutron star can be parameterized by the following equation:

$$W(E, T) = E^2 \int d\Omega\, f(\mathbf{r}, \mathbf{p}, l) \sim E^a \exp\{-(2+\alpha)E/E_{\text{av}}\} \qquad (2)$$

Here, $\Omega$ denotes the solid angle of vector z, $E_{av}$ is an average energy, and $\alpha$ is a numerical parameter describing the amount of spectral pinching; the value $\alpha = 2$ corresponds to a Maxwell–Boltzmann spectrum, and $\alpha = 2.3$ to a Fermi–Dirac distribution with zero chemical potential.

Beyond the protoneutron star surface in a neutrino sphere region, it is impossible to maintain both the chemical equilibrium between neutrinos and stellar matter and diffusion; however, a noticeable energy exchange between neutrinos and strongly magnetized stellar material can affect neutrino spectra. Since heavy-leptonic neutrinos only interact with star matter through a neutral current, they are energetically less coupled to stellar plasma than the electron flavor when neutral and charged currents are involved; therefore, the heavy-leptonic neutrinos break out of thermal equilibrium in the energy sphere, which is significantly deeper inside the nascent protoneutron star than the transport sphere (near to neutrino sphere), where the transition from diffusion to free flow breaks in. Within the scattering atmosphere, respective heavy-leptonic neutrinos still frequently collide with neutrons and protons. As is demonstrated in Sections 3.2 and 3.3, in this case, magnetic effects noticeably enhance the energy exchange in neutrino–nucleon scattering due to the neutral current.

### 3.1. Neutrino Sphere Properties

Matter in the neutrino sphere region corresponds to a moderate density $n \sim 0.1$–10 Tg cm$^{-3}$ (1 Tg = $10^{12}$ g) and temperature $T \sim 5$–10 MeV. We assume strong fluctuations

of temperature $T$ and density $n$ in this region since it meets a strong convection of matter and corresponds with the vicinity of the bifurcation point for stellar material between a collapse of the central compact object and supernova ejecta. Figure 1a shows the Fermi energy of nucleons $E_F^N$ and electrons $E_F^e$ versus a beta equilibrium parameter $Y_e$ at density $n = 1$ Tg/cm$^3$. One sees that at the realistic numbers of the beta equilibrium parameter $Y_e \sim 0.2$–0.3, these values for nucleons (protons- $E_F^p \sim 0.6$ MeV, neutrons- $E_F^n \sim 1.1$ MeV) and electrons ($E_F^e \sim 35$ MeV) are small and large when compared with temperature, respectively. Therefore, nucleon components, with $E_F^N \ll T$, represent non-degenerate gas, whereas an electron gas, with $E_F^e \gg T$, is strongly degenerated. As a consequence, a neutrino–electron scattering cross section is strongly suppressed because of the Pauli principle. Such a blocking effect also leads to the actual termination of a charged current component in neutrino–nucleon scattering. Magnetization gives rise to the effective increase in Fermi energy and further diminution of respective scattering. Moreover, the corresponding mean free path (mfp) rises up to 10ths km at considered densities; therefore, we hereafter neglect the right hand side of Equation (1). On the contrary, neutrino–nucleon scattering due to a neutral current component can be considered an independent process with corresponding mfp $l_f = (N_N \, \sigma_{GT0})^{-1} \sim 100$ m. Here, $N_i = n_i/m_i$ represents the number density of $i$-th nuclear particle (N denotes nucleon), with mass $m_i$ and contribution $n_i$ to the total mass density $n$, and $\sigma_{GT0}$ denotes the respective cross section $\sigma_{GT0} \approx 10^{-40}$ cm$^2$ $(E/37$ MeV$)^2$; see [4,5].

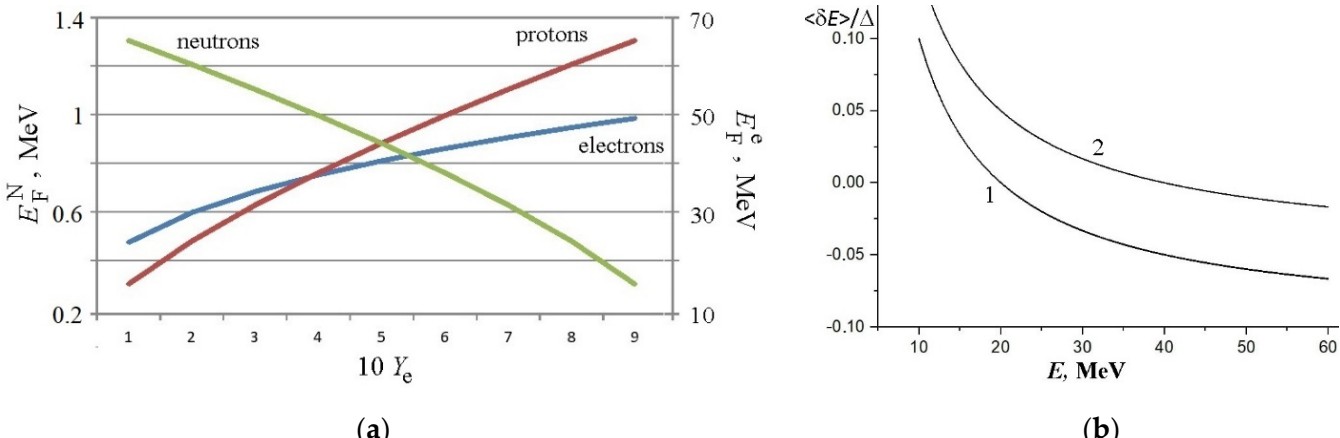

**Figure 1.** Neutrino sphere properties: (**a**) Fermi Energy at density $n = 1$ Tg cm$^{-3}$ versus beta equilibrium parameter $Y_e$. (**b**) The neutrino energy dependence of an average transferred energy $\langle\delta E\rangle$ during inelastic scattering at temperatures $T = 5$ and 10 MeV, corresponding to the curves 1 and 2. The approximate relationship in Equations (3) and (4) coincide within the line widths.

The energy transfer cross section in a magnetized neutrino sphere for $\nu + N \rightarrow \nu' + N'$ scattering has been considered by Kondratyev et al. [11]. We briefly recall that this value is defined as $S_1^i = -\int d\varepsilon \, \varepsilon \, (d\sigma_{\nu \rightarrow \nu'}^i/d\varepsilon)$ with an energy transfer $\varepsilon$ differential cross section $(d\sigma_{\nu \rightarrow \nu'}^i/d\varepsilon)$. The nucleon energy levels with spin magnetic moments directed along (spin up) and opposite (spin down) the magnetic field are split on the value $\Delta = |g_\alpha| \, \mu_N H \equiv |g_\alpha| \, \omega_L$ because of an interaction with field $H$. Here, $\mu_N$ denotes the nuclear magneton, $\omega_L = \mu_N H$ gives the Larmour frequency, and $g_\alpha$ is the nucleon g-factor. At the temperature $T$ for neutral GT0 neutrino nucleon scattering, the energy transfer cross section reads [11]

$$S_1 \approx \sigma_{GT0} \, \Delta \, (2\delta_E - (1 + \delta^2{}_E) \, \text{th}(\delta_T/2))|_{\Delta<E,T} \tag{3}$$

$$\approx \sigma_{GT0} \, \Delta \, (2\delta_E - \delta T/2) \tag{4}$$

where $\delta_E = \Delta/E$, $\delta_T = \Delta/2\,T$ and th($x$) is the hyperbolic tangent. For a magnetized nucleon gas, the average transferred energy in inelastic neutrino scattering is evaluated as being $\langle\delta E\rangle = -\langle\varepsilon\rangle \approx S_1/\sigma_{GT0}$. This value depends on temperature $T$, splitting $\Delta$ in the nucleon

gas, and the energy of the incoming neutrino $E$. Figure 1b shows the average transferred energy $<\delta E>$ for a magnetized nucleon gas in units of $\Delta$ as a function of neutrino energy $E$ at various temperatures $T$. As is seen in Figure 1b, the average transferred energy varies from a positive value for a hot nucleon gas to a negative value for a cold system. This change corresponds to a transition from exo-energetic to endo-energetic neutrino scattering, which occurs under the conditions corresponding to the temperature $T \approx E/4$, see [11]. The physical reason of such a transition is obviously because of the decreasing thermal population of the upper split energy level of the nucleons, which results in a suppression of the contribution of GT0 transitions from this level to the underlying level. The condition of such a transition from one mode to another is well described by the relation of $E \approx 4T$, which is independent on splitting value $\Delta$.

### 3.2. Energy Transfer in Neutrino Spectra

Making use of Equation (4), we determine an energy transfer coefficient as

$$\frac{\partial E}{\partial l} = \sum_i N_i S_i \approx E\left(1 - \frac{E}{4T}\right)/l_t, \tag{5}$$

where the sum goes over nuclear particles and the energy transfer length reads $l_t = E^2/(\sum_i N_i \, \sigma^i_{GT0})\Delta^2 \approx 100 \text{ m } (3 \text{ MeV}/\Delta)^2 (10 \text{ Tg cm}^{-3}/n)$. Neglecting the right hand side of Equation (1) for the uniform flux $\mathbf{z} \, \partial f/\partial \mathbf{r} = 0$, the solution of Equation (1) is given by replacing $E$ with the solution of Equation (5); see Appendix A, i.e.,

$$E_\nu \rightarrow e_l E \, (e_l + (1 - e_l) \, E/4T)^{-1} \tag{6}$$

with $e_l = \exp\{l/l_t\}$.

Figure 2 shows the energy transfer effect in neutrino energy spectra during evolution in the vicinity of the neutrinospheric region. The Maxwell–Boltzmann distribution corresponds to $\alpha = 2$ and $E_{av} = 10$ MeV in Equation (2) and is taken as the initial one. One sees that the energy transfer effect leads to an increase in neutrino energy at the maximum distribution. When neutrino path $l$ approaches a mean energy transfer length $l_t$, we obtain a spreading in distribution $W(E)$ with the maximum point increasing in a near linear manner with a growing $e_l$. Such an acceleration is particularly effective at larger gas temperatures.

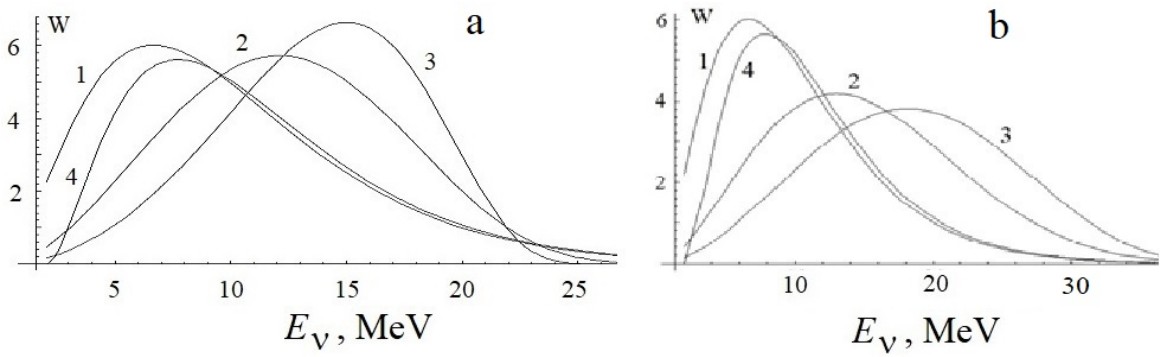

**Figure 2.** Neutrino energy spectra in arbitrary units for $\alpha = 2$, $E_{av} = 10$ MeV, and $e_l = 1$ (i.e., $l = 0$)—curves 1, 2—curves 2, and 3—curves 3 at $T =$ (**a**) 5 MeV (**b**) 10 MeV. The curves 4 correspond to a respective single effective collision event at $\Delta = 2$ MeV.

The case of a single effective collision in Figure 2 clarifies the obtained results. In this case, the relationship between corresponding exo- and endo-energetic regimes is determined by a ratio of the occupation of respective nucleon levels and neutrino phase space volume in the exit channel (i.e., $\text{Exp}\{\delta_T\} \, (1 - \delta_E)^2 \theta(1 - \delta_E)/(1 + \delta_E)^2$, with step function $\theta(x)$). When this ratio is less than 1, the number of endo-energetic collisions is larger than the number of exo-energetic collisions, and vice versa; therefore, for neutrino dynamics in magne-

tized nucleon gas, it is preferable that a change of acceleration will occur, and that the stopping regimes correspond to the condition $\delta_T = -2\ln\{(\theta(1 - \delta_E)(1 - \delta_E)/(1 + \delta_E)\}|_{\delta E, \delta T < 1} \approx 4\delta_E$. Such a switch in neutrino dynamics is also displayed in Equations (3) and (4). For large energies $E > 4T$, such effects proceed faster, compared with small energies $E < 4T$, because of the strong energy dependence of the energy transfer coefficient.

*3.3. Fluctuation Effects in Energy Spectra*

The strong convection in the vicinity of the neutrino sphere and bifurcation point gives rise to large fluctuations in the properties of respective stellar materials. We average the results of the energy spectra modification over the fluctuations. For temperature $T$, we assume a uniform distribution in a range from 5 to 10 MeV, which is independent from density fluctuations. As is seen in Figure 3a, the maximum distribution W($E$) is shifted towards larger energies, approaching the region of 10–20 MeV. The properties of such averaged energy distribution resemble the results of 10 MeV, which support an effective acceleration mechanism at higher temperatures.

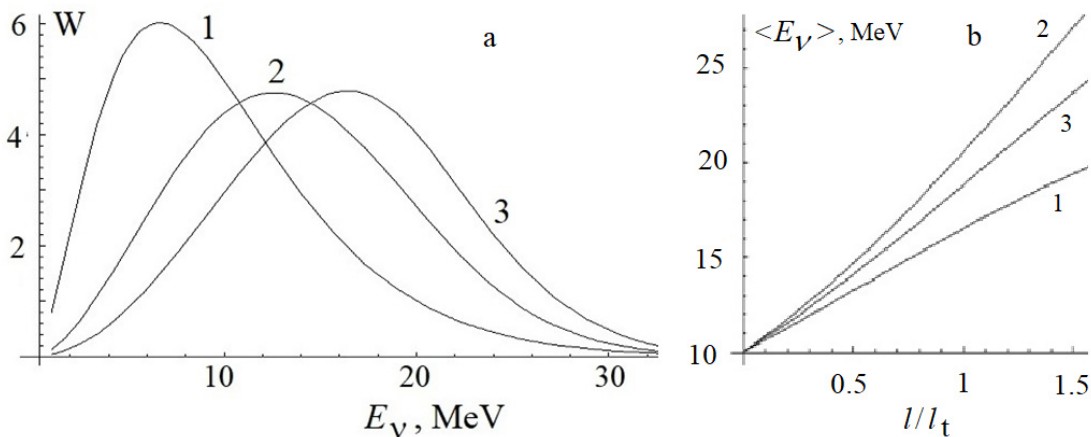

**Figure 3.** (**a**) Neutrino energy spectra in arbitrary units for $\alpha = 2$, $E_{av} = 10$ MeV, and $e_l = 1$ (i.e., $l = 0$)—curve 1, 2—curve 2, and 3—curve 3 average over $T$ from 5 MeV to 10 MeV. (**b**) The average neutrino energy depending on the path length $l$. For $\alpha = 2$, $E_{av} = 10$ MeV, $T = 5$ MeV—curve 1 and 10 MeV—curve 2. Curve 3 corresponds to the average over temperature $T$ in the range of 5 to 10 MeV.

## 4. Strengthening of Spectrum Hardness and Large Neutrino Detector Sensitivity

The spectrum hardness can be characterized quantitatively using the average energy of particles:

$$<E> = \int dE\, W(E)\, E \approx e_l\, E_{av}\, (1 + (e_l - 1)\, 3E_{av}/16T)^{-1} \tag{7}$$

In Equation (7), we used the Saddle Point method justified at $e_l \sim 1$. As can be seen in Figure 3b, the value $<E>$ increases almost linearly with the increasing path length $l$. One sees an effective energy absorption by neutrinos from magnetized stellar material. Such a feature can result in double (or multiple) peaked SN light curves. The growing hardness of neutrino energy spectra makes it a favorable feature of the considered effect for detector sensitivity.

Strongly variable transient particle fluxes can be detected using large-volume neutrino telescopes: KM3NeT [12], Baikal-GVD [13], and so on. Sensitivity to neutrinos on a scale of 10 MeV can be achieved by observing a collective increase, using multiple detectors to measure the rate of counting coincidences. A sharp increase in the spatially uniform neutrino flux $\Phi(t)$ is associated with a CCSN infall phase that occurs during the half a second [14] that defines the observation time. Assuming a spherically uniform neutrino

emission, one obtains $\Phi(t) = L(t)/4\pi d^2$ with the neutrino luminosity $L$, and the distance to the source $d$. The CCSN neutrino detection rate, $r_{SN}(t)$, can be evaluated as

$$r_{SN}(t) = \varphi \, \Phi(t) \, \Sigma_i \, n_i \int d\varepsilon \, W(\varepsilon) \, \sigma^i(\varepsilon), \tag{8}$$

where the sum index $i \in \{p, e^-, {}^{16}O\}$ represents the most important target components producing energetic charged particles (i.e., $e^{+/-}$, $n_i$ is the number of targets, $\sigma^i(\varepsilon)$ the total interaction cross section for the given target $i$, and $W(\varepsilon)$ gives the energy spectrum from Equation (3)). The detector efficiency $\varphi$ corresponds to the ratio between the number of detected events and the number of interacting neutrinos per unit of water [15].

At time $t$ of time interval $\delta t$, the probability of a triggered detector $p = r(t) \, \delta t / \Pi$, where $\Pi$ is the number of detectors and total the detection rate $r = r_{SN} + r_B$ includes the background event rate $r_B$. The multiple coincidences of $k$ detectors meet with the probability given by Poissonian law $p^k/k! \, e^{-p}$. In this case, the ratio signal/background is given by $(1 + r_{SN}/r_B)^k \approx (1 + k \, r_{SN}/r_B)$. Evidently, the $k$-fold coincidence enhances by a factor of $k$, which is the detection sensitivity for a weak SN neutrino signal. When the condition $(k \, r_{SN}/r_B) \gg 1$ corresponds to value $k$ approaching ten, then an excess of hundreds or thousands in terms of the total number of detectors $\Pi$ is required for $d \sim 10$ kpc, respectively.

## 5. Conclusions

We considered energy transfer for neutrino nuclear scattering in strong magnetic fields, which may plausibly arise in supernovae, and its respective effect in neutrino energy spectra. Nuclear magnetization is shown to bring new, neutral-current induced reaction channels, giving additional noticeable mechanisms to the dynamic of neutrinos being weakly coupled with matter. The energy transfer coefficient in kinetic equations is demonstrated to change from positive to negative values with increasing neutrino energy. For magnetized nondegenerate nucleon gas, such cross over between acceleration and stopping regimes occurs when neutrino energy is about factor four of the gas temperature, while the nucleon Larmour frequency is sufficiently small. This switching in dynamical properties originates from the detailed balance principle, and a difference of phase space volume for neutrinos in the final channel when scattering spin-up and spin-down nucleons, and independent on splitting value $\Delta$. Consequently, such a property is insensitive to the magnetization geometry. The respective acceleration and/or stopping rates are determined by the product of splitting $\Delta$ and scattering cross section $\sigma_{GT0}$ in the nucleon gas. For realistic properties of stellar material (see Section 3.1), such neutrino–nuclear scattering effects result in an increase in the hardness of neutrino energy spectra. Since electronic neutrinos decouple from matter at the neutrino-sphere, and thereafter, they experience a few (single in average) effective collisions, the corresponding acceleration effect is relatively small. Beyond the energy sphere, the dynamics of heavy-leptonic neutrinos is mainly governed by collisions with nucleons. Within the scattering atmosphere (up to the neutrino-sphere) these collisions are frequent enough to maintain spatial diffusion for heavy-leptonic neutrinos. The significant completed path $l$, within a magnetized region of a star, leads to a considerable acceleration effect in the case of a heavy-leptonic component. The strengthening of neutrino hard energies is favorable for supernova neutrino observations that use Large Volume Neutrino Telescopes. In this case, the CCSN neutrino flux is revealed from an increase in the counting rate of multiple detectors that are coincidentally triggered. As illustrated, the $k$-fold coincidence enhances the detection sensitivity for a weak SN neutrino signal by factor $k$. Finally, we notice that such strong magnetization also arises in neutron star mergers, magnetar crusts, and heavy-ion collisions.

**Author Contributions:** Conceptualization and methodology, V.N.K.; formal analysis, investigation, V.N.K., D.B.T. and T.D.L.; writing—original draft preparation, V.N.K.; writing—review and editing, V.N.K., D.B.T. and T.D.L. All authors have read and agreed to the published version of the manuscript.

**Funding:** This research received no external funding.

**Data Availability Statement:** Not applicable.

**Conflicts of Interest:** The authors declare no conflict of interest.

**Appendix A**

To find neutrino energy $E_l$ after passing a distance $l$ in neutrinospheric region we rewrite Equation (5) as

$$\mathrm{d}E_l/[E_l\,(1 - E_l/4T)] = \mathrm{d}l/l_\mathrm{t},\, E_l\,|_{\,l=0} = E \tag{A1}$$

The solution is given by

$$E_l/(E_l - 4T) = [E/(E - 4T)]\exp\{l/l_\mathrm{t}\}, \tag{A2}$$

and leads to Equation (6).

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
