# Peer review of "Effect of Protoneutron Star Magnetized Envelops in Neutrino Energy Spectra"

_2571-712X, doi:10.3390/particles5020011_

Round 1
Reviewer 1 Report
The authors study the effect of magnetized envelops on neutrino spectra for proto-neutron stars. The results are clearly presented and the manuscript is recommended for publication on Particles. Here are a few comments for the authors to improve the quality of the manuscript.
- the authors need to expand the introduction section and discuss more in detail about previous work and how the current study is better than previous ones. Right now the comparison is not clear to me
- The authors claim that the strong variable transient particle flux can be detected by KM3NET. However, an estimate of the observation time & the number of detectors is needed for collecting a reasonable number of events for statistical studies.
- Can the authors comment on the back-reaction from the neutrinos to SN gas dynamics in the presence strong magnetic field? Can this effect be detected by ground based / space telescopes?
- There are a few typos in the manuscript (e.g., the first sentence in the Neutrino kinetic section. quit -> quite).
Author Response
Dear Colleague,
Thank you very much for the report regarding the manuscript mentioned above, constrictive criticism and useful suggestions, which are accounted for in the revised version.
Specifically to your comments
Point 1: “the authors need to expand the introduction …” Response 1: The comment is taken into account. We add the discussion about previous works and an extension
Point 2: “The authors claim that the strong …”
Response 2: The comment is taken into account. We add the respective analysis in Sect. 4.
Point 3: “Can the authors comment on the back-reaction …”Response 3: The comment is taken into account. We add the discussion and analysis in Sect. 4.
Point 4: “There are a few typos in the manuscript …”
Response 4: Thank you very much for careful reading the manuscript. In the revised version the manuscript was amended as to remove typos and improve English.
Reviewer 2 Report
1, “Neutrino flux and/or magnetic pressures are considered as an additional key contribution to respective explosive shock wave formation. “
This line is very confusing, Neutrino flux & magnetic pressures are the the additional contribution, then what’s the primary contribution? “the mechanism of energy transfer ? Or “neutrino heating”? It’s better to phrase them in a clearer way.
2, “Another possible explosion mechanism is related to neutrino heating “
You may want to explain a little bit about neutrino heating here, or at least have a citation here, for “neutrino heating”
3, “Heavy leptonic neutrinos are energetically less 63 coupled to stellar plasma. “
Could you please explain it?
4, “Since this region corresponds also a bifurcation point we assume strong fluctuations of temperature T and density n in conjunction with strong convection. “
Could you rephrase this line? It’s very confusing.
5,
“ One sees that at realistic numbers of beta-equilibrium parameter Ye for nucleons and electrons these values are small and large as compared to temperature…”
You mean “small” for nucleons”, “large” for “electrons”? what are “these values”? “At” Realistic numbers? What’s the real subject here?
6, “which occurs under the condition i.e., at temperature “
Why do you put “i.e.” here?
7, “This switching in dynamical properties originates from the detailed balance principle and a difference of phase space volume for neutrino in final channel at scattering on spin-up and spin-down nucleons and independs on 186 splitting value ∆ “
“Independs”??? You mean “independent of”?
8, At realistic properties of stellar material such neutrino-nuclear scattering effects result in an increase 190 of a hardness of neutrino energy spectra.
What do you mean by saying “At realistic properties”?
9, you described the basic equations used for neutrino dynamics, but you didn’t mention your general methodology. What simulation/computation/software did you use? How do you connect your equations to your results&graphs?
Author Response
Dear Colleague,
Thank you very much for the report regarding the manuscript mentioned above, constrictive criticism and useful suggestions, which are accounted for in the revised version.
Specifically to your comments
Point 1: “Neutrino flux and/or magnetic …”
Response 1: The comment is taken into account. We add the discussion about these contributions.
Point 2: “Another possible explosion mechanism …”
Response 2: The comment is taken into account. We add explanation about neutrino heating.
Point 3: “Heavy leptonic neutrinos are energetically …”
Response 3: The comment is taken into account. We add the justification of this statement.
Point 4: “Since this region corresponds also a bifurcation …”Response 4: The comment is taken into account. This line is rephrased.
Point 5: “One sees that at realistic numbers …”
Response 5: Yes, it means “small” for nucleons”, “large” for “electrons”. These values are shown in Fig.1a. The “Realistic numbers” are given.
Point 6: “Why do you put “i.e.” here?”
Response 6: The comment is taken into account. “i.e.” is taken off.
Point 7: ““Independs”??? You mean “independent of”? …”Response 7: The comment is taken into account. The sentence is corrected.
Point 8: “What do you mean by saying “At realistic properties”? …
”Response 8: The comment is taken into account. The meaning of “realistic properties” is explained.
Point 9: “you described the basic equations used for …”? …”
Response 9: In this work we develop further simplistic scheme to evaluate an effect of hot magnetized matter on an emerging neutrino energy spectrum. The consideration is based on our previous work [10] that estimated the energy transfer cross-section of neutral-current spin flip interaction and leads to analytic results derived in Appendix. As is illustrated in the article, such an approach gives clear picture of an influence of inelastic scattering on neutrino spectra and warrants to express the publication in Particles.
Round 2
Reviewer 2 Report
Dear Authors,
Could you fix the following grammar mistakes? I strongly suggest you use some grammar checker in your editor.
1. "In vicinity of neutrino sphere strong convection -> In the vicinity of"
2. "Respectively, prompt SN explosions can be associated with magnetic pressure contributing significantly energy"
significantly-> significant
3. "Since neutrinos and/or magnetic pressure are capable to make a significant ***"
capable to -> capable of making / able to make
and/or: choose one, don’t use both
4. “Since heavy lepton neutrinos ***”
You mean "heavy-leptonic" ?
5. “versus a portion of electrons Ye at density n = 1 Tg/cm3” what do you mean a portion of? isn’t the ϒ “beta- equilibrium parameter”?
6. “On the contrary neutrino-nucleon scattering due to neutral current component can be considered as an independed process with corresponding “ independed-> independent
7. “At a temperature T for neutral GT0 neutrino nucleon scattering **”
a-> the
8. “The strong convection in vicinity of neutrino sphere and a bifurcation point ***” -> in the vicinity of
9. “For temperature T we assume a uniform distribution in a range from 5 to 10 MeV independent on density fluctuations”
“independent on” -> “ which is independent from”
10. “At realistic properties of stellar material, see section 3.1, such neutrino-nuclear scattering effects result in an increase of a hardness of neutrino energy spectra “ put “see section 3.1” in parentheses
Author Response
Dear Colleague,
Thank you very much for the report regarding the manuscript , constrictive criticism and useful suggestions, which are accounted for in the revised version.
Points 1-10: “…”
Response 1-10: The comments are taken into account. The sentences are corrected.